# Protein Fold Classification at Scale: Benchmarking and Pretraining

Dexiong Chen [* 1]   Andrei Manolache [* 2 3]   Mathias Niepert [2]   Karsten Borgwardt [1]

## Abstract

Classifying protein topology is essential for deciphering biological function, but progress is held back by the lack of large-scale benchmarks that avoid duplicates and by models that do not scale well. We introduce TEDBench, a large-scale, non-redundant benchmark for protein fold classification constructed from the Encyclopedia of Domains (TED) and Foldseek-clustered AlphaFold structures. We show that on TEDBench, current protein representation learning methods either require very large models or fail to deliver strong performance. To address this challenge, we propose Masked Invariant Autoencoders (MiAE), a self-supervised framework for protein structure representation learning. MiAE uses an extremely high masking ratio of up to $90\%$ with an $SE(3)$-invariant encoder and a lightweight decoder that reconstructs backbone coordinates from the latent representation and mask tokens. MiAE scales well and outperforms supervised counterparts and state-of-the-art baselines on TEDBench, establishing a strong recipe for protein fold classification. To test transfer beyond AlphaFold structures, we further benchmark on a curated dataset from experimental structures of CATH v4.4. TEDBench is available at https://github.com/BorgwardtLab/TEDBench.

## 1. Introduction

The recent explosion of predicted protein structures has reached a pivotal scale, with hundreds of millions of models now available through the AlphaFold Database (Jumper et al., 2021; Varadi et al., 2024). This data abundance presents a unique opportunity for protein modeling to undergo an "ImageNet moment": a transition toward large-scale, standardized supervised benchmarks that drive architectural innovation and systematic evaluation (Deng et al., 2009). However, in structural biology, the lack of widely adopted, non-redundant supervised tasks at this scale remains a bottleneck, preventing the kind of rapid iteration seen in computer vision and natural language processing.

A natural organizational unit for large-scale structural supervision is the protein domain, a modular substructure that recurs across proteins and often corresponds to coherent functional and evolutionary units (Alberts et al., 2002; Wang et al., 2021). Structural classification systems such as CATH (Orengo et al., 1997; Waman et al., 2025) arrange protein domains into a hierarchy of nested categories, capturing progressively finer-grained structural regularities from coarse shape classes to detailed evolutionary relationships. Because this hierarchy encodes meaningful structural similarity and eventually functional similarity, CATH labels have long served as reference standards for fold assignment and structural evaluation (Redfern et al., 2007b; Csaba et al., 2009; Dawson et al., 2017; Nallapareddy et al., 2023). This naturally suggests a supervised learning objective akin to object recognition: given a protein's 3D structure, predict its domain-level structural class.

Until recently, defining such an objective at scale was hindered by the difficulty of consistently segmenting domains and assigning reliable labels across millions of structures. The Encyclopedia of Domains (TED) (Lau et al., 2024) overcomes this barrier by decomposing the AlphaFold Database into domain units and mapping many of them to CATH categories using scalable, structure-based matching. By leveraging recent advances in domain segmentation (Lau et al., 2023; Wells et al., 2024; Zhu et al., 2023) and fast structural search (Van Kempen et al., 2024), TED makes large-scale structural annotation feasible for the first time.

Building on this resource, we introduce TEDBench, a large-scale benchmark for predicting CATH domain categories from protein structures. The task is formulated as a standard multi-class classification problem: given a protein structure (and sequence), predict the CATH Topology (T-level) label of its *largest* domain, defining a single, unambiguous target per structure. To reduce redundancy while preserving structural diversity, we project TED annotations onto Foldseek-clustered AlphaFold structures (Barrio-Hernandez

*Equal contribution [1]Max Planck Institute of Biochemistry, Martinsried, Germany [2]Computer Science Department, University of Stuttgart, Germany [3]Bitdefender, Romania. Correspondence to: Dexiong Chen <dchen@biochem.mpg.de>.

*Proceedings of the 43rd International Conference on Machine Learning*, Seoul, South Korea. PMLR 306, 2026. Copyright 2026 by the author(s).

et al., 2023). The resulting benchmark consists of 462,175 predicted structures and 27,638 experimental structures as an external test set, substantially exceeding the scale of existing supervised structure-based datasets, which have only tens of thousands of proteins, and complementing prior protein machine learning benchmarks that focus on functional targets.

We use TEDBench to establish baselines for representative equivariant and protein representation learning models, and to identify effective training recipes for fold classification at this scale. Existing approaches either require very large models or achieve limited performance on TEDBench. The strongest supervised-from-scratch equivariant baseline reaches only 65.44 macro-F1 on the external test set, while several widely used pretrained models fall below this under linear probing. To provide a strong reference point, we introduce Masked Invariant Autoencoders (MiAE), a self-supervised framework for learning protein structure representations from 3D coordinates. MiAE masks an extremely high fraction (up to 90%) of structural frames, each corresponding to all atomic coordinates within selected residues, and trains the model to reconstruct the masked coordinates in 3D space. Inspired by masked autoencoders for images (He et al., 2022), MiAE adopts an asymmetric encoder–decoder architecture: the encoder processes only a sparse set of visible frames, while a lightweight decoder reconstructs the full structure using the latent representation and mask tokens. This design enables efficient scaling and strong representations. On TEDBench, MiAE provides a strong reference for future methods: when trained from scratch, it improves macro-F1 by 10.23 points over equivariant baselines; after self-supervised pretraining, it achieves up to 70.44 macro-F1 under linear probing and further benefits from task-specific fine-tuning, outperforming supervised counterparts and state-of-the-art pretrained sequence and sequence–structure models despite using fewer parameters.

## 2. Related Work

Our work centers on a large-scale benchmark for protein fold classification, complemented by a self-supervised structure pretraining approach that serves as a strong reference method. We therefore review related work on fold classification, large-scale structure retrieval, geometric deep learning, and protein representation learning.

**Protein fold classification.**  Protein fold (topology) classification is a core problem in structural biology and machine learning. Structural taxonomies such as CATH (Orengo et al., 1997; Dawson et al., 2017; Waman et al., 2025) organize protein domains into hierarchical labels that capture coarse-to-fine regularities of three-dimensional structure. Within this framework, predicting a domain's topology from

its 3D structure can be formulated as a supervised recognition problem under strong geometric constraints. Previous structural benchmarks for fold classification have only up to 15K proteins (Hou et al., 2018; Rao et al., 2019; Kucera et al., 2023; Hartout et al., 2025), while TEDBench has 490K proteins, more than *30 times larger*.

**Structure comparison and large-scale retrieval.**  Historically, fold assignment relied on structure alignment and nearest-neighbor search, transferring labels from reference structures based on alignment scores (Holm & Sander, 1993; Shindyalov & Bourne, 1998; Redfern et al., 2007a). This paradigm has recently scaled to massive databases through fast structure search tools such as Foldseek (Van Kempen et al., 2024). Key enablers include high-accuracy structure predictors, most notably AlphaFold (Jumper et al., 2021), which underpin large resources such as the AFDB (Varadi et al., 2024) and the ESM Atlas generated with ESM-Fold (Lin et al., 2023), as well as standardized identifiers and curated repositories provided by UniProt (Consortium, 2024) and PDB (Berman et al., 2000).

**Geometric deep learning on molecules.**  Structural datasets have motivated geometric deep learning approaches (Bronstein et al., 2021) that operate directly on atomic or residue-level coordinates while respecting Euclidean symmetries. In particular, SE(3)-equivariant models such as MACE (Batatia et al., 2022) and GotenNet (Aykent & Xia, 2025) leverage these symmetries via message passing and perform well across molecular tasks. Frameworks like E3NN (Geiger et al., 2022) have further standardized equivariant architectures and accelerated their adoption.

**Protein representation learning.**  Beyond equivariant models, several methods learn transferable protein representations from sequence and structure. ProteinMPNN (Dauparas et al., 2022) uses message passing conditioned on backbone geometry for sequence prediction. Masked Inverse Folding (MIF) (Yang et al., 2022) combines masked language modeling with structure conditioning and benefits from transfer from large sequence-only language models. Large transformer pretraining on sequences, exemplified by ESM2 (Lin et al., 2023), has shown that sufficiently large models implicitly encode rich structural information. Subsequent works such as GearNet (Zhang et al., 2023), SaProt (Su et al., 2024b), and PST (Hartout et al., 2025) integrate structural context into protein language models.

Despite progress in structure retrieval and representation learning, it is unclear how existing models perform on a large-scale, purely supervised, non-redundant domain-level fold classification task. Using TED (Lau et al., 2024), we define a standardized large-scale prediction setting and make the following contributions: (1) *we intro-*

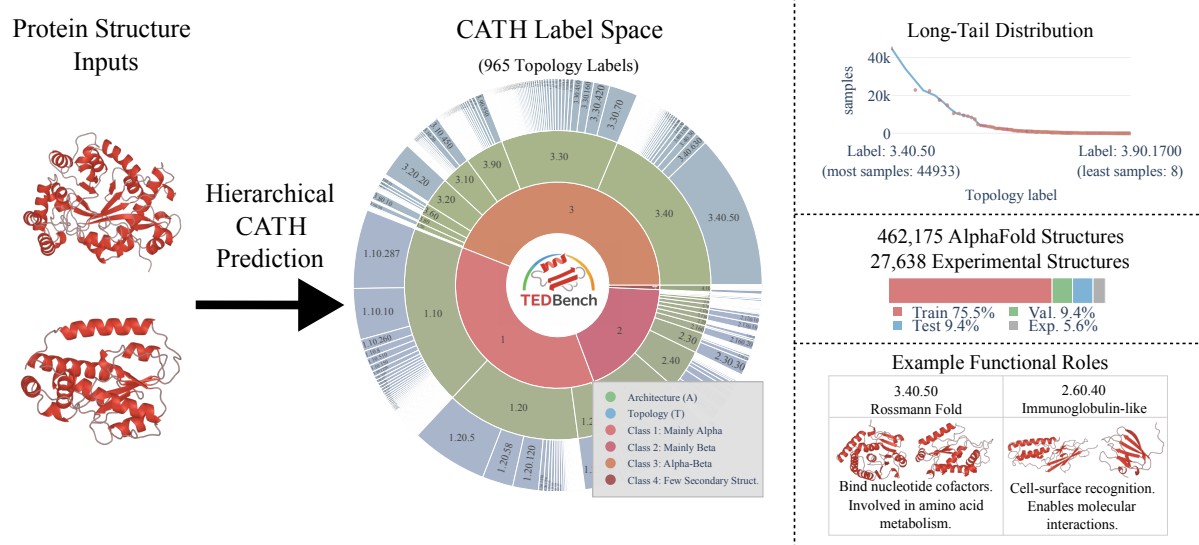

*Figure 1.* **Overview of the TEDBench.** TEDBench is a large-scale, non-redundant benchmark for protein fold classification. Given the high diversity of protein structures, CATH (Orengo et al., 1997) provides a hierarchical classification of protein domain structures. TED (Lau et al., 2024) extends this to AFDB. Our TEDBench builds upon TED and contains more than 460K predicted structures and 27K experimental structures as an external test set. Classifying protein structures offers important biological insights into their functions.

*duce TEDBench, a non-redundant benchmark for topology classification over predicted structures; (2) we propose MiAE, a lightweight self-supervised framework for learning structure-centric representations through masked autoencoding; (3) we benchmark against both general-purpose equivariant molecular models and protein-specific representation learning methods and consistently outperform them; and (4) we evaluate on an independent test set of experimental CATH structures, showing that models trained on TEDBench transfer beyond predicted structures.*

## 3. Benchmark

In this section, we detail the construction of TEDBench, a large-scale, non-redundant benchmark for protein fold classification based on domain-level structural annotations.

### 3.1. Dataset Construction

TEDBench is built upon the TED resource (Lau et al., 2024), which provides domain annotations for the entire AFDB. TED decomposes predicted protein structures into domains and assigns them to the CATH hierarchy using scalable, structure-based matching. Domain segmentation is performed using multiple algorithms, including Merizo (Lau et al., 2023), Chainsaw (Wells et al., 2024), and UniDoc (Zhu et al., 2023), with consensus filtering applied to remove low-confidence domain boundaries. Domain labels are then assigned via Foldseek and Merizo-search, enabling automated annotation at the AFDB-level scale.

While TED identifies and clusters approximately 365 mil-

lion domains across the full AFDB, this scale includes substantial structural redundancy. To construct a manageable and diverse benchmark, we restrict TEDBench to the Foldseek-clustered AFDB (Barrio-Hernandez et al., 2023), which contains representative proteins from non-singleton structural clusters. This subset comprises approximately 2.27 million proteins and substantially reduces redundancy while preserving structural diversity.

We map TED domain annotations onto this clustered AFDB subset and retain only proteins with high-confidence structural predictions (mean pLDDT > 80). The resulting TEDBench dataset contains 462,175 protein structures. Because a protein may contain multiple annotated domains, each associated with a distinct TED code, we assign each protein the CATH label corresponding to its largest domain, yielding a single, unambiguous target per structure. We also release the full unannotated clustered AFDB subset with pLDDT > 80 consisting of 749,679 protein structures, which can serve as a pretraining set.

### 3.2. Label Processing and Data Splits

A challenge in large-scale structural classification is the highly imbalanced, long-tailed distribution of CATH Topology (T-level) labels. Many T-level classes contain only a handful of examples, making reliable evaluation difficult. To mitigate this issue, we enforce a minimum class size of 10 samples by merging underrepresented T-level classes into a coarser category within the same Architecture (A-level).

Concretely, for each A-level class, all T-level classes with

fewer than 10 samples are grouped into a single aggregated label (denoted with an "x" suffix). For example, if the A-level class `1.40` contains T-level classes `1.40.10` and `1.40.20` with fewer than 10 samples each, these are merged into a new class `1.40.x`, representing other topologies within that architecture. After regrouping, the final label space consists of 965 classes, each supported by at least 10 samples.

Given that TEDBench is already non-redundant by construction, we adopt a simple random stratified split to form training, validation, and test sets. We use a ratio of $[0.8, 0.1, 0.1]$, preserving class proportions across splits. Dataset statistics are summarized in Table 1, and the resulting benchmark and class distribution are shown in Figure 1.

### 3.3. External Test Set on Experimental Structures

To assess generalization beyond predicted structures, we construct an external test set based on experimentally determined protein structures from the CATH v4.4 40% non-redundant set. This set consists of 27K proteins spanning 880 T-level classes and is fully disjoint from TEDBench. All structures are derived from experimentally resolved coordinates, providing a stringent evaluation of model robustness to domain shifts between predicted and experimental data.

*Table 1.* Statistics of TEDBench.

| | Foldseek-clustered AFDB | | | CATH v4.4 |
|---|---|---|---|---|
| | train | val | test | external test |
| # | 369,740 | 46,217 | 46,218 | 27,638 |

## 4. Masked Invariant Autoencoders

As a strong reference approach for TEDBench, we introduce Masked Invariant Autoencoders (MiAE), which extends the MAE paradigm (He et al., 2022) to the domain of 3D protein structures. MiAE adopts an asymmetric architecture where a heavy, SE(3)-invariant encoder processes only a sparse subset of visible residues, while a lightweight decoder reconstructs the full protein backbone from latent representations and mask tokens.

### 4.1. Preliminaries: Protein Frame Representation

To represent protein structures in a form amenable to geometric learning, we model each residue as a local coordinate frame. A frame encodes both the position and orientation of a rigid body in 3D space and provides a compact, rotation-aware representation of local structure. Following prior work (Ingraham et al., 2019; Hayes et al., 2025), we associate each residue $i$ with a transformation $\mathbf{T}_i \in$ SE(3),

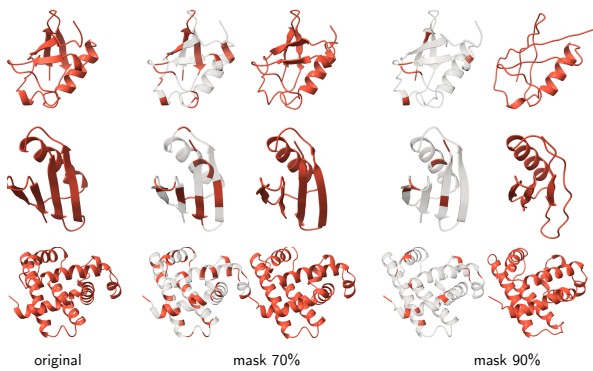

*Figure 2.* Reconstructions of *experimental structures* using an MiAE pre-trained with a masking ratio of 90%. The predictions recover well the shape and secondary structures, even when using a high masking ratio. Masked residues are masked in gray.

represented as a $4 \times 4$ homogeneous matrix:

$$\mathbf{T}_i = \begin{bmatrix} \mathbf{R}_i & \mathbf{t}_i \\ 0_{1\times 3} & 1 \end{bmatrix} \in \text{SE}(3),$$

where $\mathbf{R}_i \in$ SO(3) is a rotation matrix and $\mathbf{t}_i \in \mathbb{R}^3$ is a translation vector. The translation $\mathbf{t}_i$ corresponds to the global coordinates of the residue's $\alpha$-carbon ($C_\alpha$). The rotation $\mathbf{R}_i$ defines the local orientation of the residue and is constructed from an orthonormal basis derived from the backbone atoms $(N, C_\alpha, C)$. This construction aligns the residue's local coordinate system with the protein backbone, yielding a representation that is invariant to global rigid-body transformations.

Using this formulation, points can be transformed between local and global coordinate systems as

- *Local to global:* $p_{\text{global}} = \mathbf{R}_i p_{\text{local}} + \mathbf{t}_i$,
- *Global to local:* $p_{\text{local}} = \mathbf{R}_i^\top (p_{\text{global}} - \mathbf{t}_i)$.

A protein structure is thus represented as a sequence of residue frames, which serve as the fundamental input tokens to MiAE.

### 4.2. Masked Invariant Autoencoders

MiAE follows the MAE paradigm, adapted to 3D protein geometry and SE(3)-invariant representations (Figure 3).

**Masking strategy.** Given a protein represented as a sequence of residue frames, we randomly sample a subset of frames to retain and mask the remaining ones. Frames are sampled uniformly without replacement, and the masking ratio, defined as the fraction of removed frames, is typically very high (up to 90%). Such aggressive masking substantially reduces local redundancy and prevents trivial reconstruction via interpolation from neighboring residues. Uniform random sampling avoids introducing structural biases and produces highly sparse inputs, which in turn enables an efficient encoder design.

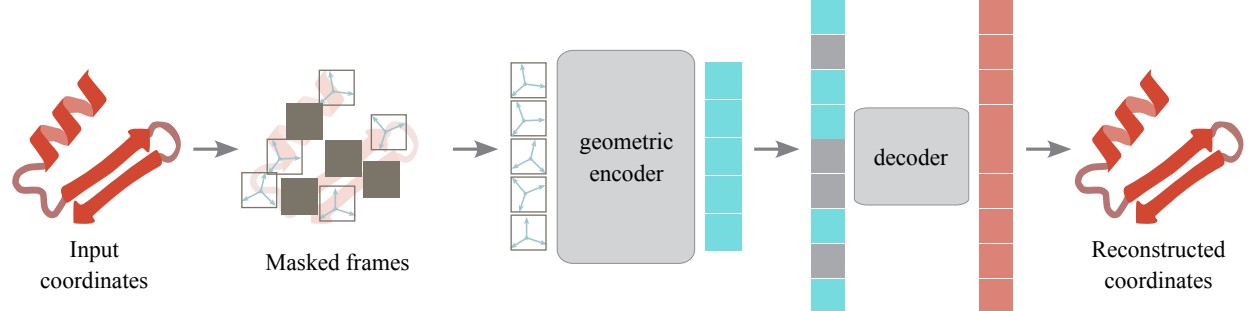

*Figure 3.* **Overview of the MiAE architecture.** During pre-training, a high masking ratio (*e.g.,* 90%) is applied to backbone frames. The geometric encoder processes only this small subset of unmasked frames, maintaining SE(3)-invariance relative to the input coordinates. Following the encoder, mask tokens are reintegrated into the latent sequence. A lightweight decoder then operates on the full set of encoded frames and mask tokens to reconstruct the original backbone coordinates. After pre-training, the decoder is discarded, and the encoder is applied to uncorrupted backbone coordinates for downstream tasks.

**MiAE geometric encoder.** The encoder processes only the visible frames and consists of two geometric attention blocks followed by a standard Transformer encoder. Each geometric block comprises a geometric self-attention layer and a feed-forward layer, following the design of ESM3 ([Hayes et al., 2025](#)). Unlike ESM3, where geometric attention is restricted to the $k$ nearest neighbors of each residue, we apply attention globally across all visible frames. Because the visible set is small (*e.g.,* 10% of residues), this global attention remains computationally efficient.

Masked frames are entirely removed at this stage; no mask tokens are introduced in the encoder. This design allows the encoder to scale to large model sizes while operating on only a fraction of the full sequence. After the geometric blocks, positional embeddings are added, and the resulting representations are passed through a standard Transformer encoder. Further details of the geometric self-attention mechanism are provided in Appendix B.1.

**MiAE decoder.** The decoder receives the full sequence of tokens, consisting of (i) encoded visible-frame representations and (ii) learned mask tokens corresponding to the missing frames. Each mask token is a shared embedding that indicates a residue whose structure must be reconstructed. The decoder comprises a small number of Transformer blocks with rotary positional embeddings ([Su et al., 2024a](#)).

The decoder is used exclusively during pre-training and is discarded afterward. As a result, its architecture can be designed independently of the encoder. In practice, we employ decoders that are significantly shallower and narrower than the encoder, with less than 10% of the per-token computational cost. This asymmetric design ensures that the full sequence is only processed by a lightweight network, substantially reducing pre-training time.

**Reconstruction target.** MiAE is trained using the composite reconstruction loss $\mathcal{L}_{\text{ESM3}}$ introduced in ESM3 ([Hayes et al., 2025](#)). This objective combines five terms, with geometric distance and geometric direction losses serving as the primary supervision signals for accurate backbone reconstruction. Auxiliary binned distance and direction classification losses help stabilize training, while an inverse folding token prediction loss encourages representations that are informative for sequence-related tasks. We ablate the effect of the inverse folding loss in our experiments. Note that, unlike MAE, our loss is operating on all backbone atoms instead of masked atoms. Further details on the loss functions are provided in Appendix B.2.

**Incorporating amino acid sequence.** MiAE can optionally incorporate amino acid sequence information. When enabled, amino acids corresponding to masked frames are also masked, while the remaining residues are embedded and added to the encoded visible frame representations alongside positional embeddings. This additional signal improves fold classification performance in our experiments.

## 5. Experiments

We evaluate a diverse set of baselines and MiAE on TED-Bench. Our experiments are designed to answer three questions: (i) how challenging is TEDBench and how do existing methods perform; (ii) does MiAE learn transferable structure representations; and (iii) which design choices are critical to MiAE's performance on TEDBench. We further analyze MiAE's latent space through qualitative visualization.

### 5.1. Experimental Setup

Due to the highly imbalanced distribution of fold classes, we report macro F1 score in addition to accuracy on both the test and external test set (see Section 3). Macro F1 better

*Table 2.* **Benchmark results on TEDBench. Bold** and underline represent the best and second-best overall results.

| model | size | test | | external test | |
|---|---|---|---|---|---|
| | | acc | F1 | acc | F1 |
| *supervised from scratch* | | | | | |
| GotenNet | 1.9M | 73.33 | 64.02 | 82.61 | 65.44 |
| E3NN | 1.9M | 71.87 | 57.63 | 71.15 | 42.40 |
| MACE | 1.5M | 66.89 | 50.58 | 69.73 | 44.73 |
| MiAE-S | 29M | 78.05 | 70.03 | 88.87 | 74.38 |
| MiAE-B | 102M | **78.36** | 71.60 | 89.06 | 75.02 |
| MiAE-B+seq | 102M | 78.23 | **71.64** | **89.40** | **75.67** |
| MiAE-L | 339M | 78.34 | 70.95 | 89.17 | 75.03 |
| *pretrained + finetuned* | | | | | |
| ESM2-35M | 35M | 68.14 | 46.47 | 83.01 | 58.65 |
| ESM2-150M | 150M | 73.26 | 57.07 | 86.21 | 65.85 |
| ESM2-650M | 650M | 76.63 | 66.19 | 88.59 | 72.29 |
| SaProt-35M | 35M | 79.46 | 71.75 | 89.22 | 74.75 |
| SaProt-650M | 650M | **80.31** | 73.48 | **90.22** | 76.78 |
| MiAE-S | 29M | 79.15 | 72.28 | 89.60 | 76.08 |
| MiAE-B | 102M | 79.90 | 73.71 | 89.84 | 75.72 |
| MiAE-B+seq | 102M | **80.31** | **74.56** | 90.08 | **77.34** |
| MiAE-L | 339M | 80.10 | 73.47 | 90.01 | 76.46 |

reflects per-class performance and is our primary metric.

We consider three training protocols: *supervised training from scratch*, *linear probing*, and *fine-tuning*. Supervised training refers to models trained directly on TEDBench without pretraining. In this setting, we evaluate several state-of-the-art general-purpose $E(3)$-equivariant molecular models, including GotenNet (Aykent & Xia, 2025), E3NN (Geiger et al., 2022), and MACE (Batatia et al., 2022), as well as variants of MiAE (encoder) trained from scratch.

Linear probing and fine-tuning are applied to pretrained models. For linear probing, the pretrained encoder is frozen and a linear classifier is trained on top. We include publicly available pretrained models spanning different input modalities: structure-only models such as ProteinMPNN (PMPNN Dauparas et al. (2022)) and MIF (Yang et al., 2022); sequence-only models including variants of ESM2 (Lin et al., 2023); and sequence–structure hybrid models such as SaProt (Su et al., 2024b). For fine-tuning, we perform end-to-end training on TEDBench. Due to memory constraints, we exclude the largest ESM2 variants.

Our MiAE models are pretrained on the Foldseek-clustered, high-confidence (pLDDT$> 80$) structure subset described in Section 3. We emphasize that these pretrained models differ in architecture size and pretraining data, and comparisons should therefore be interpreted with this context in mind.

We evaluate three MiAE variants of increasing capacity: MiAE-S (29M parameters/6 layers), MiAE-B (102M/12L),

and MiAE-L (339M/24L). All models are optimized using AdamW with a cosine learning rate schedule. For fine-tuning, we apply layer-wise learning rate decay. Additional implementation details are provided in Appendix B.3.

## 5.2. Benchmark Results on TEDBench

Table 2 summarizes the main benchmark results on TED-Bench. Among supervised-from-scratch baselines, the strongest general-purpose molecular model (GotenNet) achieves a macro F1 score of $64.02$. In contrast, all MiAE variants substantially outperform these baselines, with even the smallest MiAE-S exceeding GotenNet by over 6 points. This gap highlights the difficulty of TEDBench and suggests that effective fold classification requires structure-aware models with sufficient capacity.

Most methods exhibit strong positive transfer to the external test set derived from CATH v4.4, suggesting that learned representations transfer well from AFDB-predicted structures to experimentally resolved ones. Notably, MiAE models achieve approximately $10\%$ higher accuracy on the external test set than on the test split. We attribute this to the limited diversity of experimental structures compared to AlphaFold-predicted ones and the higher label fidelity of human-curated CATH annotations. However, scaling MiAE from MiAE-B to MiAE-L yields no or only marginal gains, and incorporating sequence information does not consistently improve performance in this setting.

When fine-tuning, MiAE obtains further gains. The MiAE+seq variant achieves the best F1 score on both test sets, outperforming larger ESM2 and SaProt models. At a lower parameter budget level (30M), MiAE-S also outperforms ESM2-35M and SaProt-35M in F1 on both test sets. Sequence-only models such as ESM2 are generally less competitive than structure-based or hybrid approaches, which is expected given that CATH topology labels are defined directly by 3D structure. Across all MiAE variants, fine-tuning yields consistent improvements of nearly $4\%$ over training from scratch, mirroring the gains observed with MAE-style pretraining in computer vision.

Beyond performance gains, we further compare the computational time of ESM2, SaProt and MiAE for both pre-training and fine-tuning in Appendix C.1. Our comparison suggests that MiAE requires much fewer computational resources to achieve similar or better performance than state-of-the-art protein representation learning models.

## 5.3. Representation Quality of MiAE

**Linear probing.** To evaluate representation quality, we freeze the MiAE encoder and train a linear classifier on top of mean-pooled residue embeddings (other pooling strategies are ablated in Appendix C.3). Results are reported in

*Table 3.* Linear probing results on TEDBench. **Bold** and underline represent the best and second-best overall results. We isolate models with more than 1B parameters.

| model | size | test | | external test | |
|---|---|---|---|---|---|
| | | acc | F1 | acc | F1 |
| *pretrained + linear probing* | | | | | |
| PMPNN | 1.6M | 54.25 | 41.43 | 59.88 | 38.92 |
| MIF | 3.4M | 56.49 | 44.38 | 52.02 | 34.36 |
| ESM2-35M | 35M | 65.71 | 41.91 | 79.48 | 52.66 |
| ESM2-150M | 150M | 70.81 | 54.39 | 84.71 | 63.36 |
| ESM2-650M | 650M | 74.65 | 62.32 | 87.15 | 70.03 |
| SaProt-35M | 35M | 73.06 | 58.82 | 85.49 | 67.08 |
| SaProt-650M | 650M | 75.23 | **66.55** | **87.31** | **70.79** |
| MiAE-S | 29M | 67.90 | 49.43 | 80.35 | 59.03 |
| MiAE-B | 102M | 72.94 | 58.52 | 85.31 | 66.18 |
| MiAE-B+seq | 102M | 74.08 | 62.14 | 86.20 | 68.88 |
| MiAE-L | 339M | **75.42** | 63.50 | 87.06 | 70.44 |
| ESM2-3B | 3B | **76.44** | 69.08 | 88.82 | 75.75 |
| ESM2-15B | 15B | 76.32 | **70.85** | **88.92** | **76.27** |

Table 3. MiAE outperforms inverse-folding-based structure models such as PMPNN and MIF. While the highest-performing model overall is ESM2-15B, it contains orders of magnitude more parameters than MiAE-L. At comparable parameter budgets ($\leq$650M), MiAE-L surpasses ESM2-650M and performs competitively with SaProt-650M.

Scaling MiAE from MiAE-S to MiAE-L yields a clear and consistent improvement, with macro F1 increasing from 59 to 70. This indicates that MiAE representations already separate fold classes effectively without task-specific adaptation.

**Fine-tuning.** We next fine-tune all models end-to-end. As shown in Table 2, fine-tuning consistently improves performance across all methods. Notably, MiAE benefits substantially more from fine-tuning than ESM2 or SaProt. For example, the gap between linear probing and fine-tuning for MiAE-B+seq reaches nearly 12.5 and 8.5 F1 points on the test and external test sets, respectively, compared to approximately 4/2 for ESM2 and 7/6 for SaProt, showing that MiAE adapts efficiently to fold classification.

### 5.4. Main Properties of MiAE

We ablate key design choices of MiAE using the default configuration from Table 2.

**Masking ratio.** Figure 4 examines the effect of the masking ratio. Consistent with MAE for images (He et al., 2022), performance peaks at very high masking ratios, up to 90%, exceeding the optimal ratio typically reported for vision models (75%) and in contrast to the typical small ratio (15%)

used in BERT-style models for texts (Devlin et al., 2019) and protein sequences (Rives et al., 2021; Lin et al., 2023).

Even pretrained with masking ratios as high as 70%, the backbone reconstruction RMSD remains low (0.57), indicating that local structural fragments are highly correlated and can be reliably interpolated. When the masking ratio exceeds 90%, RMSD increases sharply, suggesting that the model must rely on more global structural reasoning, which in turn leads to more informative representations. Visualization of some reconstructions is provided in Figure 2 and more examples in Appendix C.5, showing that MiAE still infers plausible structures even when using a very high masking ratio. This high masking result reflects that protein backbones possess significant *local redundancy* and *recurring structural motifs*, consistent with biological research into the "tertiary alphabet" and reusable structural motifs (Mackenzie et al., 2016).

When pretrained with a standard autoencoder, *i.e.,* with a masking ratio of 0.0, we observe a sharp performance drop in both linear probing and fine-tuning (Table 4). This suggests that without the challenge of reconstruction from sparse inputs, the model fails to learn the global structural features necessary for fold classification. Beyond performance, the high masking ratio allows the heavy encoder to operate on only 10% of the residues, drastically reducing computational overhead. Note that the fine-tuned non-masking MiAE performed slightly worse than the one trained from scratch (Table 2). This is likely because the fine-tuning protocol uses layer-wise learning rate decay and fewer epochs, optimized for the MAE paradigm, which may lead to sub-optimal convergence if the initial weights are not sufficiently good.

*Table 4.* **Masking vs non-masking.** Macro F1 scores on both the test/external test set are provided.

| masking ratio | linear probing | fine-tuning |
|---|---|---|
| 0.9 (default) | 58.52/66.18 | 73.71/75.72 |
| 0.0 (non-masking) | 45.70/23.90 | 71.41/74.36 |

**Decoder design.** We study the impact of decoder depth and width in Tables 5a and 5b. Increasing decoder depth improves performance when mean pooling is used, but can degrade performance when relying on the `[CLS]` token. Mean pooling generally delivers higher performance than the `[CLS]` token; we use mean pooling as our default setting. The increased performance when using a deep decoder can be explained similarly to MAE: the last several layers in an autoencoder are more specialized for reconstruction, but are less relevant for recognition. A reasonably deep decoder can account for the reconstruction specialization, leaving the latent representations at a more abstract level.

Decoder width has a pronounced effect, and we find that

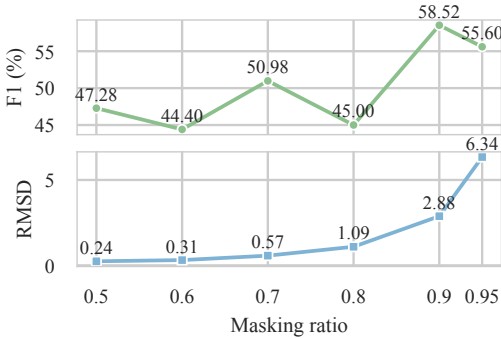

*Figure 4.* **Masking ratio.** A high masking ratio tends to deliver higher linear probing performance and higher reconstruction error (RMSD). The test performance is plotted.

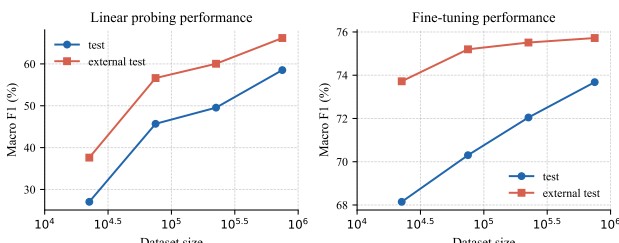

*Figure 5.* **Pretraining data size.** MiAE-B achieves better downstream performance with an increasing amount of pretraining data.

a width of 512 provides the best performance in our setting. However, this optimal value could vary when using a decoder depth, and we recommend users to jointly tune decoder depth and width carefully in practice, even though the decoder is discarded after pretraining.

**Reconstruction loss.** We ablate the inverse folding loss term included in the reconstruction objective (Section 4.2). While the other terms are all subject to the structural reconstruction of the protein, this is the only loss term related to the sequence. Removal leads to a clear drop in performance, confirming that sequence-level supervision encourages the latent representations to retain information useful for downstream tasks. Note that in contrast to the original MAE for computer vision, our loss operates on *all backbone atoms* instead of only masked atoms, as pairwise distances and directions are used to maintain the SE(3)-invariance.

**Masking strategy.** We compare two different mask sampling strategies: random masking and random span masking.

The default random masking strategy randomly samples a subset of frames to mask without replacement. Random span masking samples contiguous masks for a given length (5) and complements the masks with a random sampling strategy if it does not reach the required mask length. This strategy is more challenging and mimics tasks like motif

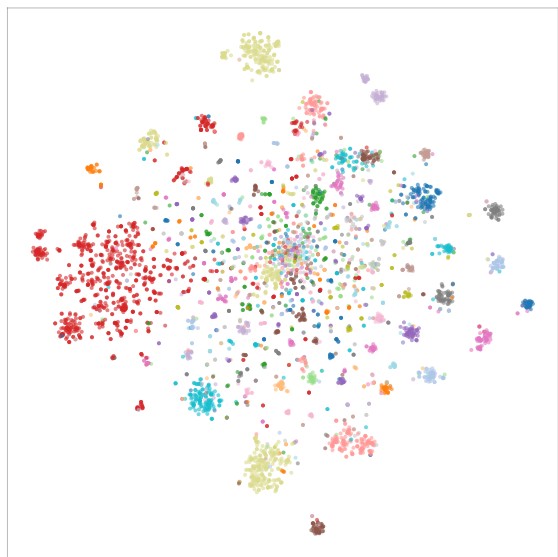

*Figure 6.* t-SNE projection of pretrained MiAE protein embeddings (before fine-tuning), colored by CATH topology; several topologies form clear neighborhoods in the learned space.

scaffolding. Table 5e shows that random span masking works slightly better than random masking while being more complicated. Therefore, we keep the simpler random masking strategy as our default setting.

**Scaling and sequence incorporation.** MiAE scales effectively with model size under linear probing, with MiAE-L outperforming MiAE-S by over 15 points in F1 (Table 5c), though its benefits are less pronounced under full fine-tuning (Table 2). Furthermore, increasing the amount of pretraining data also improves downstream performance for MiAE-B in both linear probing and fine-tuning regimes (Figure 5). Incorporating amino acid sequence information further improves linear probing (Table 5f) and fine-tuning performance (Table 2).

### 5.5. Latent Space and Attention Visualization

To provide qualitative intuition for what MiAE learns before any supervised fine-tuning, we visualize the pretrained representation space with t-SNE (van der Maaten & Hinton, 2008). We encode each protein with the pretrained MiAE encoder and form a single protein-level embedding by mean-pooling the final-layer residue representations. Figure 6 shows the 2D projection colored by CATH topology, while Supplementary Figure 8 includes additional views colored by class and architecture. Across these views, the embeddings exhibit a meaningful organization with respect to the CATH hierarchy: in particular, class and architecture labels tend to occupy distinct regions of the map even

*Table 5.* **MiAE ablation experiments** with linear probing on TEDBench. We report both accuracy (acc) and macro F1 score (F1). If not specified, the default setting (marked in gray) is MiAE-B with a decoder of depth 2 and width 512, the reconstruction loss is the composite loss $\mathcal{L}_{ESM3}$, including the inverse folding loss (invf), the masking strategy is random, and the masking ratio is 90%.

| blocks | avg | cls |
|---|---|---|
| 1 | 55.65 | 46.61 |
| 2 | 58.52 | 34.74 |
| 4 | 59.65 | 13.26 |

*(a)* **Decoder depth.** A deep decoder can lead to gains for avg but not cls pool.

| dim | acc | F1 |
|---|---|---|
| 256 | 61.62 | 35.50 |
| 512 | 72.94 | 58.52 |
| 768 | 54.48 | 27.83 |

*(b)* **Decoder width.** The decoder needs to be narrower than the encoder.

| size | acc | F1 |
|---|---|---|
| MiAE-S | 67.90 | 49.43 |
| MiAE-B | 72.94 | 58.52 |
| MiAE-L | 75.42 | 63.50 |

*(c)* **Model size.** Larger models achieve better linear probing performance.

| case | acc | F1 |
|---|---|---|
| w/ invf | 72.94 | 58.52 |
| w/o invf | 70.47 | 52.55 |

*(d)* **Reconstruction loss.** Inverse folding loss is useful when added to the objective.

| case | acc | F1 |
|---|---|---|
| random | 72.94 | 58.52 |
| span | 73.35 | 59.23 |

*(e)* **Masking strategy.** Span masking is slightly better, but more complicated.

| case | acc | F1 |
|---|---|---|
| w/o seq | 72.94 | 58.52 |
| w/ seq | 74.08 | 62.14 |

*(f)* **Sequence incorporation.** Incorporating amino acid sequence is useful.

when they do not form tight clusters, while topology reveals finer-grained neighborhoods within and across those regions. We include these visualizations as a qualitative perspective on the learned space, complementing our probing and fine-tuning experiments.

We also visualize the attention weights of an end-to-end fine-tuned MiAE model in Appendix C.2. These plots reveal the structural components prioritized by the model, offering interpretability with potential biological applications.

# 6. Conclusion

We introduced TEDBench, a large-scale supervised benchmark for protein fold classification. TEDBench provides a challenging yet well-defined setting for evaluating structure-based representation learning methods at scale. We complemented this benchmark with MiAE, a simple self-supervised reference approach for protein structure representation learning that scales well and performs strongly on TEDBench. MiAE is SE(3)-invariant and shares properties similar to masked autoencoders for computer vision, combining high optimal masking ratios with an asymmetric encoder–decoder architecture.

We hope that TEDBench will serve as a standardized evaluation platform for future work on large-scale protein structure modeling. More broadly, MiAE demonstrates that masked autoencoding principles can be successfully extended to protein geometry, opening avenues for scalable self-supervised learning across a range of structural biology tasks.

**Limitations.** This work has several limitations. First, TEDBench focuses on coarse-grained protein-level fold recognition. Similar to computer vision, this task could be further reformulated as a detection problem by training models to simultaneously segment and classify domains. Because TEDBench assigns each protein the label of its largest annotated domain, it can miss smaller domains that may be biologically relevant; future work can instead perform domain-level segmentation and classify the resulting domains individually. Second, while MiAE demonstrates strong performance for fold classification, we have not explored its transferability to tasks beyond structural categorization, such as function prediction or interaction modeling. Finally, although MiAE is computationally efficient during pretraining due to aggressive masking, training large-scale models still requires substantial resources, which may limit accessibility. Addressing these limitations is an important direction for future work.

# Acknowledgements

The authors would like to thank the anonymous reviewers for their insightful feedback. AM and MN acknowledge the support from the International Max Planck Research School for Intelligent Systems (IMPRS-IS). AM acknowledges funding by the EU Horizon project ELIAS (No. 101120237).

# Impact Statement

From a societal perspective, improved protein structure modeling could indirectly support applications in drug discovery, enzyme design, and synthetic biology. These domains have clear positive potential, such as enabling more targeted therapeutics or environmentally sustainable biochemical processes. At the same time, advances in protein modeling also raise ethical considerations related to dual-use risks, as similar techniques could be repurposed to assist in the design of harmful biological agents. While MiAE focuses on representation learning rather than generative design or sequence synthesis, and operates on backbone structures rather than full biochemical pipelines, we acknowledge that progress in foundational modeling can lower barriers for

downstream misuse if integrated into broader systems.

This work does not involve human subjects, personal data, or sensitive biological datasets. All training data are derived from publicly available protein structures, and the method does not aim to predict or infer biological function beyond structural representations. Nevertheless, responsible deployment of models built on top of MiAE should follow established best practices in biosecurity, including controlled access, careful evaluation of downstream applications, and alignment with community guidelines for responsible AI in the life sciences.

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

# Appendix

## A. Data Processing Details of TEDBench

The full TED annotations are downloaded from https://zenodo.org/records/13908086 and the list of representative proteins in the FoldSeek cluster is downloaded from https://afdb-cluster.steineggerlab.workers.dev. The TED domain annotations are mapped onto this subset filtered with high-confidence structural predictions (mean pLDDT > 80). As described in Section 3.2, all T-level classes with fewer than 10 samples are grouped into a single aggregated label. This is implemented as follows:

```python
def process_cath_labels(cath_codes, cts_cutoff=20):
    """
    Process CATH labels by aggregating rare topology classes.

    T-level (topology) classes with fewer than cts_cutoff samples are
    merged into an aggregated label within their parent A-level
    (architecture). For example, rare topologies "1.40.10" and "1.40.20"
    are merged into "1.40.x".

    Args:
        cath_codes: List of CATH codes in C.A.T or C.A.T.H format
        cts_cutoff: Minimum samples required to retain a T-level class

    Returns:
        labels: Unique CATH labels after aggregation
        label_indices: Index mapping from input codes to labels
        label_counts: Sample count per label
    """
    # Normalize codes to C.A.T format (add .x if only C.A level)
    normalized_codes = []
    for code in cath_codes:
        if code.count('.') == 2:
            code += '.x'
        normalized_codes.append(code)

    # Extract C.A.T level (ignore H-level if present)
    T_codes = ['.'.join(code.split('.')[:3]) for code in normalized_codes]

    # Get unique T-level labels and their counts
    T_labels, T_inv, T_cts = np.unique(
        T_codes, return_inverse=True, return_counts=True
    )

    # Group T-level classes by their A-level parent (C.A)
    A_groups = {}
    for i, t_code in enumerate(T_labels):
        a_code = '.'.join(t_code.split('.')[:2])  # Extract C.A
        A_groups.setdefault(a_code, []).append(i)

    # Determine which T-level classes should be merged
    merge_mask = np.zeros(len(T_labels), dtype=bool)

    for a_code, t_indices in A_groups.items():
        t_indices = np.array(t_indices)
        counts = T_cts[t_indices]
        rare_mask = counts < cts_cutoff

        if rare_mask.sum() == 0:
```

```
            continue

    # If total rare samples < cutoff, include one more class
    if counts[rare_mask].sum() < cts_cutoff:
        # Include the smallest non-rare class
        safe_counts = np.where(rare_mask, np.inf, counts)
        rare_mask[safe_counts.argmin()] = True

    merge_mask[t_indices[rare_mask]] = True

# Apply merging: rare T-codes become C.A.x
processed_codes = [
    '.'.join(T_codes[i].split('.')[:2]) + '.x' if merge_mask[T_inv[i]]
    else T_codes[i]
    for i in range(len(T_codes))
]

# Return final unique labels and mappings
labels, label_indices, label_counts = np.unique(
    processed_codes, return_inverse=True, return_counts=True
)

return labels, label_indices, label_counts
```

For experimental structures of CATH v4.4, we downloaded the 40% non-redundant set from `ftp://orengoftp.biochem.ucl.ac.uk/cath/releases/latest-release/` and processed their annotations using the label mapping precomputed on the above dataset.

## B. Implementation Details

### B.1. MiAE Geometric Encoder

The MiAE encoder adopts the geometric self-attention mechanism introduced in ESM3 (Section A.1.6.2 and Algorithm 6 in Hayes et al., 2025). In contrast to standard self-attention, which operates solely on per-residue embeddings, geometric attention additionally incorporates per-residue rigid frames $T$, enabling the integration of structural information in a rotation- and translation-invariant manner.

Our implementation closely follows ESM3, with two notable simplifications. First, whereas ESM3 employs relative positional embeddings within each $k$-nearest-neighbor neighborhood, MiAE uses a single learned embedding shared across all residues. In this way, the self-attention is applied globally rather than within nearest neighborhoods. Second, absolute sinusoidal positional embeddings are not injected into the geometric attention blocks; instead, they are added only after the geometric encoder stack. These design choices simplify the architecture while preserving the inductive bias of geometry-aware attention.

### B.2. MiAE Reconstruction Target

MiAE is trained end-to-end using a composite reconstruction objective following ESM3 (Hayes et al., 2025). The overall loss is defined as

$$\mathcal{L}_{\text{ESM3}} = \mathcal{L}_{\text{dist}} + \mathcal{L}_{\text{dir}} + \mathcal{L}_{\text{binned dist}} + \mathcal{L}_{\text{binned dir}} + \mathcal{L}_{\text{inverse folding}}. \tag{1}$$

The continuous distance and direction losses ($\mathcal{L}_{\text{dist}}$, $\mathcal{L}_{\text{dir}}$) provide the primary supervision signal for backbone reconstruction. The binned classification losses ($\mathcal{L}_{\text{binned dist}}$, $\mathcal{L}_{\text{binned dir}}$) act as auxiliary objectives to stabilize and bootstrap training. Finally, an inverse folding loss encourages the learned representations to retain sequence-level information relevant to downstream tasks. Pairwise logits for the classification losses are produced following Algorithm 9 in ESM3.

**Backbone distance loss $\mathcal{L}_{\text{dist}}$.** This loss penalizes discrepancies between predicted and ground-truth pairwise backbone distances. For each structure, we compute the pairwise $L_2$ distance matrices over the three backbone atoms $(N, C_\alpha, C)$ for both predicted and true coordinates, yielding $D_{\text{pred}}, D \in \mathbb{R}^{3|V| \times 3|V|}$. The loss is defined as

$$\mathcal{L}_{\text{dist}} = \text{mean}(\min((D_{\text{pred}} - D)^2, 25)),$$

where the truncation mitigates the influence of large outliers.

**Backbone direction loss $\mathcal{L}_{\mathbf{dir}}$.** The direction loss captures relative orientation information between residues. For each residue, we compute six vectors from both predicted and ground-truth coordinates: (a) $N \rightarrow C_\alpha$; (b) $C_\alpha \rightarrow C$; (c) $C \rightarrow N_{\mathrm{next}}$; (d) $-(N \rightarrow C_\alpha) \times (C_\alpha \rightarrow C)$; (e) $(C_{\mathrm{prev}} \rightarrow N) \times (N \rightarrow C_\alpha)$; (f) $(C_\alpha \rightarrow C) \times (C \rightarrow N_{\mathrm{next}})$. Then, it computes the pairwise dot product between these vectors for both predicted and ground truth coordinates, denoted as $D_{\mathrm{pred}}, D \in \mathbb{R}^{6|V| \times 6|V|}$. Finally, we compute

$$\mathcal{L}_{\mathrm{dir}} = \mathrm{mean}(\min((D_{\mathrm{pred}} - D)^2, 20))$$

**Binned distance classification loss $\mathcal{L}_{\mathbf{binned\ dist}}$.** This auxiliary loss discretizes pairwise distances to provide coarse structural supervision. Ground-truth $C_\beta$ positions are first reconstructed from $(N, C_\alpha, C)$ coordinates. Pairwise $C_\beta$ distances are then binned into $64$ intervals with lower bounds $[0, 2.3125^2, (2.3125 + 0.3075)^2, \ldots, 21.6875^2]$, yielding labels $y \in 0, \ldots, 63^{|V| \times |V|}$. Cross-entropy loss is computed between these labels and the corresponding pairwise logits, which are computed using the final decoder layer representations.

**Binned direction classification loss $\mathcal{L}_{\mathbf{binned\ dir}}$.** Similar to the above loss, this loss captures a coarser similarity between ground truth and predicted orientations to stabilize early training. Specifically, we compute the pairwise dot product between three vectors $C_\alpha \rightarrow C$, $C_\alpha \rightarrow N$, and $(C_\alpha \rightarrow C) \times (C_\alpha \rightarrow N)$ normalized to unit length. Then, we bin these dot products into 16 evenly spaced bins in $[-1, 1]$, forming classification labels $y \in \{0, \ldots, 15\}^{|V| \times |V|}$. Finally, we compute the pairwise logits as above and compute the cross-entropy loss using the labels $y$ and the logits.

**Inverse folding loss $\mathcal{L}_{\mathbf{inverse\ folding}}$.** To encourage sequence-awareness, the final decoder representations are passed through a classification head to predict amino acid identities. Given ground-truth residue types as labels, we apply a standard cross-entropy loss over the amino acid vocabulary.

### B.3. Training Details of MiAE

**Architecture.** We consider three variants of MiAE: MiAE-S, MiAE-B, and MiAE-L. We extract features from the encoder

*Table 6.* MiAE variants.

|  | # params. | # layer | hidden dim | # attn heads |
|---|---|---|---|---|
| MiAE-S | 29M | 6 | 512 | 8 |
| MiAE-B | 102M | 12 | 768 | 12 |
| MiAE-L | 339M | 24 | 1024 | 16 |

output for finetuning and linear probing. Like MAE for computer vision, in our MiAE pre-training, we append an auxiliary dummy token to the encoder input after the geometric blocks. This token will be treated as the class token for training the classifier in fine-tuning. For linear probing, we found that the model works better with average pooling.

**Supervised training from scratch.** Our hyperparameter choices largely follow those of MAE for images. Without much tuning efforts, our MiAE works well on TEDBench. We use a simple perturbation data augmentation following ProteinMPNN, by adding a small Gaussian noise with a standard deviation of $0.2$ to the input training coordinates. This slightly improved the validation performance by $0.6\%$. We provide our recipe in Table 7a.

**Pretraining.** We did not perform any hyperparameter tuning for optimization and simply followed the same recipe from MAE (He et al., 2022). We provide the hyperparameters in Table 7c.

**Linear probing and end-to-end fine-tuning.** For linear probing, as the encoder is frozen, we first extract protein representations and normalize them using a standard scaler. Then, a linear classifier is trained on top of them using an L-BFGS algorithm. We select the regularization parameter based on the validation accuracy in the $[100, 10, 1, 0.1, 0.01, 0.001]$.

For end-to-end fine-tuning, the training procedure is very similar to that of supervised training from scratch, except that we use a layer-wise learning rate decay (Clark et al., 2020) following MAE (He et al., 2022), and a smaller batch size, thus fewer epochs. Table 7b summarizes the hyperparameters.

*Table 7.* **Training settings of MiAE**.

*(a)* Supervised training setting of the MiAE encoder.

| config | value |
| --- | --- |
| optimizer | AdamW |
| learning rate | 0.0016 |
| weight decay | 0.1 |
| optimizer momentum | $0.9, 0.95$ |
| effective batch size | 4096 |
| learning rate schedule | cosine decay |
| warmup iterations | 1830 (about 20 epochs) |
| training iterations | 18300 (about 200 epochs) |
| data augmentation | $\mathcal{N}(0, 0.2^2)$ |

*(b)* End-to-end fine-tuning setting of MiAE.

| config | value |
| --- | --- |
| optimizer | AdamW |
| learning rate | 0.0016 |
| layer-wise lr decay | 0.8 |
| weight decay | 0.1 |
| optimizer momentum | $0.9, 0.95$ |
| effective batch size | 1024 |
| learning rate schedule | cosine decay |
| warmup iterations | 1830 (about 5 epochs) |
| training iterations | 18300 (about 50 epochs) |
| data augmentation | $\mathcal{N}(0, 0.2^2)$ |

*(c)* Pretraining setting of MiAE.

| config | value |
| --- | --- |
| optimizer | AdamW |
| learning rate | 0.0024 |
| weight decay | 0.05 |
| optimizer momentum | $0.9, 0.95$ |
| effective batch size | 4096 |
| learning rate schedule | cosine decay |
| warmup iterations | 5000 |
| training iterations | 100000 |

## B.4. Training Details of Other Baselines

We provide training details of all the baselines.

**Generic equivariant models on molecules.** We train all three equivariant baselines from scratch and attach a linear classification head on top of the mean-pooled representations. All models use an 8Å distance cutoff to construct neighborhoods. We optimize with Muon (Jordan et al., 2024) for 2D weight tensors and AdamW (Kingma & Ba, 2015; Loshchilov & Hutter, 2019) for all remaining parameters; we observe that Muon substantially improves convergence for these equivariant models. We reduce the learning rate on plateau. We set model capacities as follows: GotenNet with 4 interaction blocks and hidden size 128; MACE with 2 layers and hidden size 256; and a custom E3NN model with 10 layers and hidden size 128.

**ProteinMPNN and MIF.** We use the official checkpoint version `v_48_020` of ProteinMPNN (Dauparas et al., 2022) to extract protein structure representations from all backbone atoms (`ca_only=False`). For MIF (Yang et al., 2022), we use the model weights from the Python `sequence-models` package[1].

**ESM2.** We use the official ESM2 checkpoints from the `fair-esm` Git repository, including 5 variants: `esm2_t12_35M_UR50D`, `esm2_t30_150M_UR50D`, `esm2_t33_650M_UR50D`, `esm2_t36_3B_UR50D`, and `esm2_t48_15B_UR50D`. We use all models for linear probing and only the 3 smaller models for fine-tuning due to the high computational costs of larger models. The protein-level representations are computed through average pooling over amino acid representations, as suggested by the official tutorials. For end-to-end fine-tuning, we similarly use a layer-wise learning decay and use exactly the same hyperparameters as used for fine-tuning MiAE.

---

[1] https://github.com/microsoft/protein-sequence-models

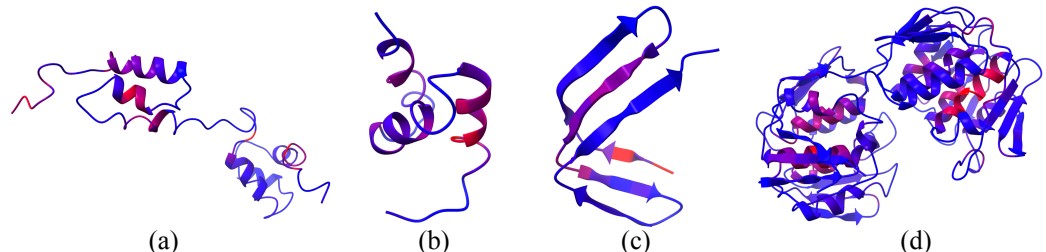

*Figure 7.* **Visualization of attention weights.** Protein samples with colored residues based on a heatmap defined by the average attention weights of the last layer of an end-to-end fine-tuned MiAE-B model. (a) and (b): two examples from the mainly alpha class; (b): mainly beta; (c): alpha and beta. The model appears to identify the core structural components.

**SaProt.** We use the official checkpoints from SaProt: `SaProt_35M_AF2` and `SaProt_650M_AF2`. We use exactly the same settings for SaProt as for ESM2.

## C. Additional Results

### C.1. Computational Time Comparison

We compare the computational time of ESM2, SaProt and MiAE for both pretraining and fine-tuning in Table 8. Our comparison suggests that MiAE requires much fewer computational resources to achieve similar or better performance than state-of-the-art protein representation learning models such as ESM2 or SaProt.

*Table 8.* Computational time comparison of ESM2, SaProt and MiAE for both pretraining and fine-tuning in GPU hours. Note that the architectures of SaProt and ESM2 and the sequence lengths are identical. The pretraining time of ESM2 and SaProt is the same and was estimated based on the numbers reported in the SaProt paper (Su et al., 2024b), while the fine-tuning results were obtained from our compute cluster using the same hardware for all models.

| Model | Pretraining | Fine-tuning |
|---|---|---|
| ESM2-650M | 138,240 | 132 |
| SaProt-650M | 138,240 | 132 |
| MiAE | 768 | 43 |

### C.2. Visualizing Attention Weights

We visualize the attention weights by the end-to-end fine-tuned MiAE-B model in Figure 7. The model appears to identify the core components contributing to the fold classification task: for alpha helix-abundant structures, MiAE focuses on residues composing alpha helices. For beta sheet-rich structures, MiAE identifies the sheet determinant for the fold class. For proteins with multiple domains, MiAE learns to look at all domains.

### C.3. Complete Ablation Results

We provide complete ablation results in Table 9 for linear probing with both average embeddings and `[CLS]` token embeddings. The performance of using average embeddings is generally superior to that of `[CLS]` token embeddings.

### C.4. Additional Latent Visualizations

Additional visualizations for Class and Architecture labels are available in Figure 8.

### C.5. Additional Reconstruction Examples

We provide additional reconstruction samples in Figure 9.

*Table 9.* Linear probing performance for MiAE with average pooling (avg) or [CLS] token representation (cls).

| masking ratio | decoder depth | decoder width | w/ invf | mask strategy | w/ seq | acc/F1 (avg) | acc/F1 (cls) |
|---|---|---|---|---|---|---|---|
| 0.9 | 2 | 512 | yes | random | no | 72.94/58.52 | 65.85/34.74 |
| 0.95 | | | | | | 70.67/55.60 | 64.84/43.98 |
| 0.8 | | | | | | 65.02/45.00 | 61.59/36.89 |
| 0.7 | | | | | | 68.14/50.98 | 59.97/30.39 |
| 0.6 | | | | | | 62.45/44.40 | 35.78/ 4.12 |
| 0.5 | | | | | | 64.44/47.28 | 58.54/ 36.53 |
| | 1 | | | | | 71.92/55.65 | 69.86/46.61 |
| | 4 | | | | | 73.89/59.65 | 52.36/13.26 |
| | | 256 | | | | 61.62/35.50 | 51.09/22.48 |
| | | 768 | | | | 54.48/27.83 | 50.40/21.03 |
| | | | no | | | 70.47/52.55 | 67.57/43.54 |
| | | | | span | | 73.35/59.23 | 70.52/47.87 |
| | | | | | yes | 74.08/62.14 | 70.46/48.00 |

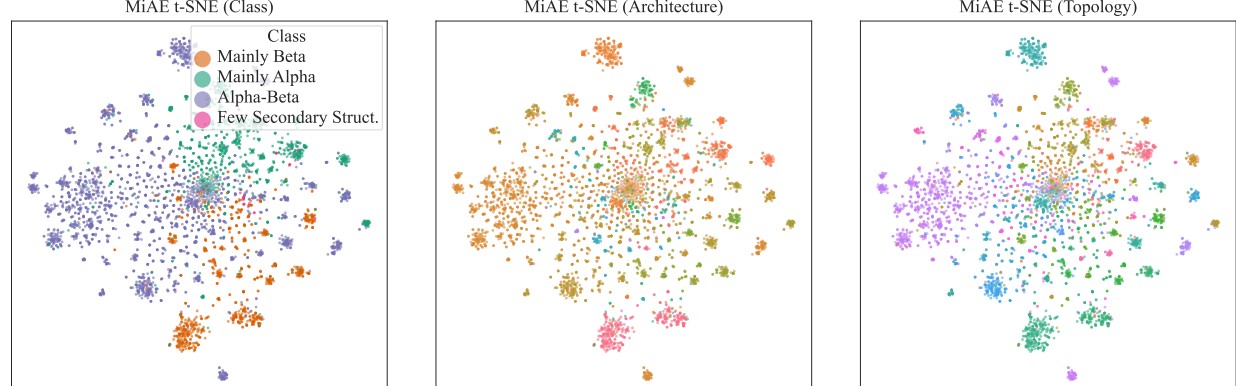

MiAE t-SNE (Class)     MiAE t-SNE (Architecture)     MiAE t-SNE (Topology)

Class
Mainly Beta
Mainly Alpha
Alpha-Beta
Few Secondary Struct.

*Figure 8.* **t-SNE projection** of protein-level embeddings produced by the MiAE encoder before fine-tuning (mean pooled from final-layer residue representations). Points correspond to proteins and are colored by CATH labels (class, architecture, and topology). The map shows that class and architecture labels tend to occupy distinct regions, while topology provides a finer-grained view with multiple topologies forming recognizable neighborhoods within and across those regions.

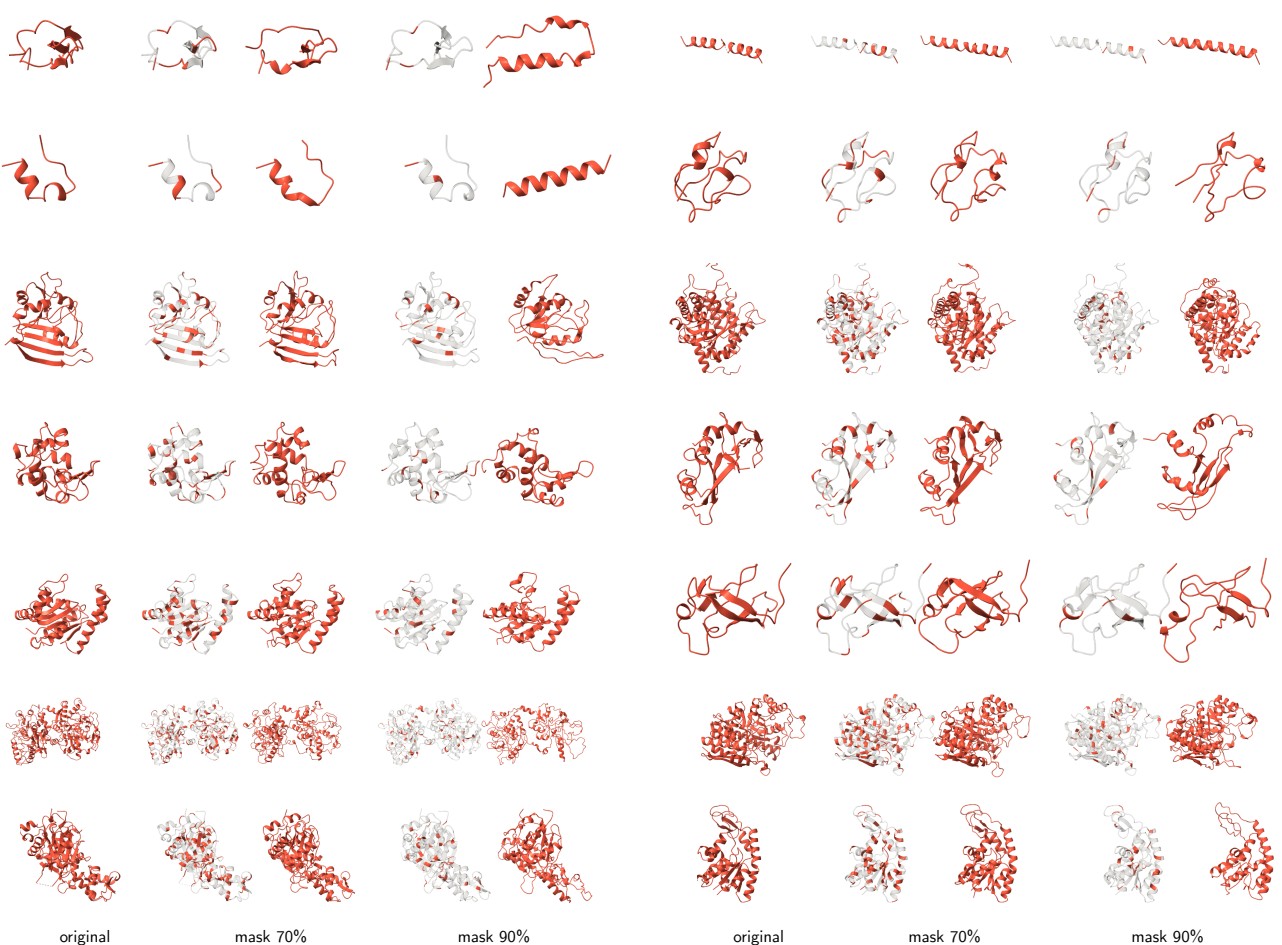

original     mask 70%     mask 90%       original     mask 70%     mask 90%

*Figure 9.* **Uncurated random samples** on the external experimental structures, using an MiAE pretrained with a masking ratio of 90% on the FoldSeek clustered dataset. For each sample, we show the original structure, the masked structure, and the reconstructed structure for two masking ratios 70% and 90%. Masked residues are marked in gray.

