# OpenReview forum: "Protein Fold Classification at Scale: Benchmarking and Pretraining"
_ICML.cc/2026/Conference — ICML 2026 spotlight_

### Official Review · Reviewer_1BK7 · 2026-03-10

**Soundness:** 3
**Presentation:** 3
**Significance:** 2
**Originality:** 2
**Overall Recommendation:** 4
**Confidence:** 4

**Summary:**

The paper introduces TEDBench, a large-scale, non-redundant benchmark for protein fold classification (CATH topology prediction), derived from the Encyclopedia of Domains (TED) and Foldseek-clustered AlphaFold structures. It comprises over 460k predicted structures and an external test set of 27k experimental structures. To address the challenge of fold classification at this scale, the authors propose masked invariant autoencoder (MiAE), a self-supervised method that applies a high masking ratio (up to 90%) to protein backbone coordinates. The model utilizes an SE(3)-invariant geometric encoder and a lightweight decoder to reconstruct the full backbone. Experiments show that MiAE outperforms several general-purpose equivariant models (e.g., E3NN, MACE) and protein language models (e.g., ESM2, SaProt) on the proposed TEDBench.

**Compliance With Llm Reviewing Policy:**

Affirmed.

**Final Justification:**

The rebuttal has fully resolved my concerns. Thus, I would like to raise my score to 4.

**Key Questions For Authors:**

1.  **Generalization to other tasks:** Can the authors provide results on other standard protein understanding benchmarks (e.g., EC prediction, GO term prediction, or binding affinity) to demonstrate that MiAE learns transferable features beyond just topology classification? This is standard for representation learning papers.
2.  **Impact of Masking vs. Architecture:** In Table 2, "Supervised from scratch" MiAE-B achieves 71.60 F1, while "Pretrained + finetuned" MiAE-B achieves 73.68 F1. The gain from pre-training is relatively modest (~2%). Does a standard Autoencoder (reconstruction without masking) yield similar gains? This would clarify if the "Masking" is essential or if the gain is simply from unsupervised initialization on more data.
3.  **Computational Cost:** How does the training cost (GPU hours) of MiAE compare to the baselines like SaProt or ESM2? The paper claims efficiency due to the lightweight decoder and sparse encoder, but concrete numbers would strengthen this claim.
4.  **Comparison with Foldseek Search:** Since the dataset is constructed using Foldseek, a strong baseline would be a k-NN classifier based on Foldseek alignment scores. How does MiAE compare to simply retrieving the nearest neighbor in the training set using Foldseek?

**Limitations:**

The authors discuss limitations regarding the focus on coarse-grained recognition and lack of functional prediction transferability in the conclusion. However, they do not adequately address the limitation that their "pre-training" might not generalize to non-structural tasks (like sequence-function mapping) given the purely geometric nature of the input. The societal impact section is generic.

**Strengths And Weaknesses:**

### Strengths

*   **Introduction of a Standardized Benchmark:** The proposal of TEDBench is the strongest contribution of this work. The field lacks a large-scale, clean, and non-redundant benchmark for structural fold classification that incorporates the scale of AlphaFold Database (AFDB) while maintaining experimental rigor (CATH labels). The data processing steps—filtering by pLDDT, handling class imbalance via aggregation, and using a distinct experimental test set—are well-motivated and sound.
*   **Effective Method for the Specific Task:** The proposed MiAE, despite its simplicity, demonstrates strong performance on the fold classification task, outperforming both supervised baselines and several pre-trained models. The finding that an extremely high masking ratio (90%) is viable for protein structure reconstruction is an interesting empirical observation.
*   **Comprehensive Baseline Comparison:** The authors compare against a wide range of baselines, including sequence-only models (ESM2), structure-only models (MACE, E3NN, ProteinMPNN), and hybrid models (SaProt), providing a good landscape of current capabilities on this specific task.

### Weaknesses

*   **Insufficient Evaluation of "Representation Learning":** The title and abstract frame this work as "Protein Fold Classification at Scale: Benchmarking and Pretraining" and claim to propose a framework for "protein structure representation learning." However, a core tenet of representation learning is that the learned features should generalize to diverse downstream tasks. The paper only evaluates the model on the fold classification task itself (which is the task the benchmark is built for). Compared to prior works like SaProt, GearNet, or ESM, which evaluate on Enzyme Commission (EC) number prediction, Gene Ontology (GO) term prediction, binding affinity, or mutation effect prediction, the evaluation here is extremely narrow. Without these tasks, it is impossible to judge if MiAE learns a general protein representation or simply overfits to the geometric features required for fold recognition. This severely undermines the "ImageNet moment" analogy used in the introduction, as ImageNet models are famous for transferring to detection and segmentation, not just classification.
*   **Missing Key Ablations on the Pre-training Objective:** While the authors ablate the masking ratio (Figure 4), there is a critical missing baseline: a standard Autoencoder (0% masking) or a Denoising Autoencoder with low noise. The paper claims the success is due to the Masked Autoencoding paradigm. However, without comparing to a non-masked version (where the encoder sees the full structure) or a version with very low corruption, it is unclear whether the performance gain comes from the masking strategy or simply from the SE(3)-invariant geometric encoder architecture itself. If a supervised model trained from scratch with the same encoder (0% mask) performs similarly to the pre-trained & fine-tuned model, then the contribution of the "Masked" pre-training is negligible. The current "supervised from scratch" baseline in Table 2 (MiAE-S/B/L) essentially serves this role, but a direct comparison of the reconstruction objective without masking is missing to justify the specific MAE design choice.
*   **Disconnect between Pre-training and Fine-tuning:** The paper emphasizes the reconstruction of local coordinates. However, fold classification is a global property. While high masking forces some global context learning, the lack of contrastive objectives or global matching objectives might limit its semantic understanding compared to models like CLIP or sequence-structure contrastive methods. The paper would benefit from discussing why reconstruction is the optimal pre-training task for fold classification specifically.

---

> ### Author Rebuttal · Authors · 2026-03-30
>
> We thank the reviewer for identifying TEDBench as a strong contribution, highlighting the simplicity and effectiveness of MiAE, and recognizing our comprehensive baseline comparison.
>
> > W1+Q1+Limitation. Insufficient evaluation of "representation learning"
>
> We think this concern is mainly about framing rather than the core contribution. The goal of this paper is not to claim a universal protein representation model across all downstream tasks, but to study protein fold classification at scale and introduce a pretraining recipe that is effective in that setting. That is also how the paper is positioned in the title and introduction, where the emphasis is on a large-scale fold classification benchmark together with a structure-centric pretraining method for this task.
>
> That said, we agree the current wording can be tightened. In the revision, we will further narrow the scope around large-scale fold classification and soften the broader “ImageNet moment” language. Evaluation on functional prediction tasks would certainly be interesting, but we view that as follow-up work beyond the scope of this paper, and we already note transfer beyond structural categorization as a limitation and future direction (see Sec. 6).
>
> > W2+Q2. Non-masking experiment
>
> We conducted an ablation study using a standard autoencoder (0% masking) for pretraining.
>
> | Masking ratio     | Linear probing (test/ext. test) | Fine-tuning (text/ext. test) |
> | ----------------- | ------------------------------- | ---------------------------- |
> | 0.9 (default)     | 58.52/66.18                     | 73.68/75.72                  |
> | 0.0 (non-masking) | 45.70/23.90                     | 71.41/74.36                  |
>
> As the results demonstrate, pretraining without masking significantly degrades performance in both linear probing and fine-tuning modes.
> - The sharp drop in linear probing for the 0.0 ratio suggests that without the challenge of reconstruction from sparse inputs, the model fails to learn the global structural features necessary for fold classification.
> - The non-masking MiAE performed slightly worse than the model trained from scratch. This is likely because the fine-tuning protocol uses layer-wise learning rate decay and fewer epochs, optimized for the MAE paradigm, which may lead to sub-optimal convergence if the initial weights are not sufficiently good.
> - We emphasize that a ~2% gain in macro-F1 is substantial given the highly imbalanced, long-tailed distribution of the labels. Improving this metric requires accurately classifying multiple rare fold classes. Furthermore, this improvement magnitude is similar to the gains seen in the original MAE for images (e.g., 82.5% vs 84.9%).
>
> Beyond performance, the 90% masking ratio is central to our claim of **scalability**; it allows the heavy encoder to operate on **only 10% of the residues, drastically reducing computational overhead** while forcing the model to learn the global structural context.
>
> > W3. Disconnect between pre-training and fine-tuning
>
> We do not claim that reconstruction is the uniquely optimal pre-training objective for fold classification, and we agree that this can be discussed more clearly. Our motivation is that fold classification is fundamentally a geometric task, so reconstructing masked backbone frames is a direct and **scalable** way to train the model to capture the 3D organization of the protein. In MiAE, the encoder sees only a sparse subset of residues, applies global geometric attention over the visible frames, and is trained to reconstruct the full backbone, so the objective is not purely local even though the supervision is given at the coordinate level. We also agree that contrastive or global matching objectives could be interesting alternatives, but we view them as complementary directions rather than necessary to show that reconstruction is effective for this setting. We will discuss the connection between our pre-training objective and the fold classification task more clearly in the revision.
>
> > Q3. Computational cost
>
> Please see the response to W1 of reviewer jSdk.
>
> > Q4. Comparison with Foldseek search
>
> Because TED labels were assigned via **Foldseek structural matching** against a reference subset of CATH, a retrieval-based baseline would be circular. Note that this approach requires domain expertise with careful hyperparameter selection and is very time-consuming, needing several months of computation.
> Our goal with MiAE and TEDBench is to move beyond the complex, multi-month annotation pipelines to a **single parametric model**.
>
> We also note that we already compared against SaProt, the most relevant learned baseline that directly incorporates **Foldseek-derived 3Di structural tokens**. MiAE outperforms SaProt on the test set (74.56 vs. 73.48), validating that our continuous coordinate representation is more effective than Foldseek-based features.

---

> > ### Author Rebuttal · Reviewer_1BK7 · 2026-04-03
> >
> > The authors have adequately addressed all my concerns in the rebuttal.

---

> > > ### Author Response · Authors · 2026-04-03
> > >
> > > Dear Reviewer 1BK7,
> > >
> > > We sincerely appreciate your positive follow-up and your confirmation that our rebuttal has addressed all your concerns. We will carefully incorporate the discussed improvements into the revision.
> > >
> > > Best regards,
> > >
> > > The Authors

---

### Official Review · Reviewer_7wxf · 2026-03-13

**Soundness:** 2
**Presentation:** 3
**Significance:** 2
**Originality:** 3
**Overall Recommendation:** 4
**Confidence:** 4

**Summary:**

In this work, the authors make two main contributions: 1. Develop a new dataset, TEDBench, which is a reduction of the TED dataset more suited for deep learning, and 2. Demonstrate masked invariant autoencoder, a masked autoencoder framework for protein structure representations, and demonstrate its strong performance on protein fold type.

The authors also rigorously compare MiAE against a wide range of baselines, including equivariant models, ESM models, and other recent approaches like ProteinMPNN and SaProt. The detailed probing and visualizations provided significantly enhance understanding of MiAE’s capabilities and limitations.

**Compliance With Llm Reviewing Policy:**

Affirmed.

**Final Justification:**

The authors addressed some of the main concerns.

**Key Questions For Authors:**

1. Biological Interpretation of High Masking: A very high masking ratio (up to 90%) was used, and still able to predict the structure. While computationally interesting, what biological insights, if any, explain this?
   What aspects of protein structure are the model learning to reconstruct despite such extensive masking? Is there any evidence suggesting the model isn’t simply learning to “fill in the gaps” without meaningful structural understanding?
2. Do the results differ for the structure with experimental evidence versus one derived only from AFDB ?

**Limitations:**

yes

**Strengths And Weaknesses:**

Strengths:

- This work is important in addressing the problem of predicting fold/topology classification, which lacks some standardized benchmarks. The TEDBench dataset is derived from TED by removing duplicates and reducing it to 450k proteins, making it a large dataset.

- Comprehensive Experimental evaluations: The authors evaluate their method on a diverse set of datasets, compare it with a substantial number of baselines, and conduct thorough ablation studies.

- The external test set uses experimentally determined CATH v4.4 structures and is fully disjoint from TEDBench, which strengthens the empirical story beyond AlphaFold-only evaluation.

- The paper is generally well-organized and clearly written, making it easy to follow the methodology and results. The supplemental material is particularly helpful, containing further details on training procedures and additional results.

- The authors acknowledge and address practical considerations like computational cost when comparing to larger models (ESM2).


Weakness

- Overall methods are reasonable, but there are some concerns about dataset creation:
  Proteins often do have multiple domains, and CATH itself is built around assigning labels to individual domains, not whole proteins. CATH explicitly notes that classification is performed on individual protein domains and that multidomain protein structures are divided into constituent domains before classification. In the paper, the dataset construction also states that “a protein may contain multiple annotated domains,” but then collapses each protein to the label of its largest domain, resulting in a single target per structure. So, important smaller domains may be ignored. If a smaller domain is functionally decisive, regulatory, or interaction-mediating, the benchmark will not reward models for recognizing it. This means the task may under-measure biologically relevant structure understanding. The paper’s own construction acknowledges multiple annotated domains per protein before selecting only the largest one.

- Some of the architectural improvements in this work has been previously used. MAE already introduced the asymmetric encoder-decoder with masking of most inputs and reconstruction from visible tokens. ESM3 already used SE(3)-invariant geometric attention over residue frames for protein structure reasoning. What seems new here is the specific combination: a protein-structure masked autoencoder with an SE(3)-invariant encoder, a very high masking ratio of up to 90%, a lightweight decoder, and a reconstruction target tailored to back plus inverse-folding supervision.

- A very high level of masking is performed, upto 90%. While this is challenging and surprising from ML perspective, it is not clear what is being learning from biological perspective.

---

> ### Author Rebuttal · Authors · 2026-03-30
>
> We thank the reviewer for highlighting the importance of addressing fold classification at scale, comprehensive experimental evaluation, and the value of our experimental test set. We address the reviewer's concerns point-by-point below.
>
> > W1. Domain collapsing
>
> We agree with the reviewer that collapsing a multi-domain protein to the label of its largest domain is a simplification. We made this choice to define a single, unambiguous target per structure for a large-scale benchmark, but it can miss smaller domains that may still be biologically important. This limitation is already noted in the paper (see Sec. 6), where we discuss moving from protein-level fold recognition toward joint domain segmentation and classification.
>
> At the same time, TED itself is built from domain-level segmentation and annotations, so this does not prevent domain-centric use in future work. A natural extension is to first segment a protein into domains using domain segmentation tools like Chainsaw, Merizo or Unidoc as TED, and then apply the classifier to the extracted domains rather than assigning a single label to the full protein. We will make this motivation and limitation clearer in the revision.
>
> > W2. Novelty
>
> We agree that MiAE builds on ideas from MAE and recent invariant protein structure models. Our claim is not that each ingredient is new in isolation, but that the full combination is nontrivial and particularly well matched to large-scale fold classification. Specifically, we wish to clarify its unique technical contributions:
>
> - Unlike prior works like ESM3 that rely on structural tokenization, MiAE is, to our best knowledge, the first to mask and reconstruct **continuous 3D backbone coordinates** directly from latent representations.
> - While ESM3 restricts geometric attention to $k$-nearest neighbors, MiAE applies attention globally across all visible frames.
> - By combining global attention with an aggressive **90% masking ratio**, we prevent the model from performing simple local interpolation and force it to learn the fundamental "grammar" of global protein topology. We believe this finding itself is novel and insightful.
>
> Beyond the method, TEDBench provides the community with a necessary, non-redundant resource for "ImageNet-scale" supervised benchmarking. This dataset fills a critical gap in evaluating structural models on diverse, human-curated topology labels at scale.
>
> > W3+Q1. Biological interpretation of high masking
>
> We do not interpret the 90% masking result as evidence that protein structure is trivial; rather, it reflects that protein backbones possess significant **local redundancy** and **recurring structural motifs**. Specifically,
>
> - At 70% masking, reconstruction remains relatively "easy" with a low backbone RMSD of 0.57, suggesting the model can interpolate fragments from nearby structural context. Our best downstream performance is reached at ratios around 90%, where reconstruction error increases sharply.  This indicates that while lower masking can be solved via local structural redundancy, 90% masking forces the model to rely on **global structural context**, which appears to produce more informative representations for fold classification.
> - This interpretation is consistent with biological research into the "tertiary alphabet" and reusable structural motifs (Mackenzie et al., 2016). We will cite this work in our revision to further discuss how the inherent degeneracy of protein architecture makes aggressive masking a requirement for learning non-trivial global features.
>
> Mackenzie et al., "Tertiary alphabet for the observable protein structural universe", PNAS 2016
>
> > Q2. Experimental vs. predicted structures
>
> Yes. We explicitly evaluate this in our external test set, which consists of 27,638 experimentally determined CATH v4.4 structures that are fully disjoint from TEDBench, whose training data is derived from AFDB/TED. Overall, the results show good transfer from predicted to experimental structures rather than a drop in performance. For example, MiAE-B+seq improves from 74.56 Macro F1 on the TEDBench test set to 77.34 on the external experimental test set after fine-tuning, and MiAE-L improves from 63.50 to 70.44 under linear probing.
>
> We will make this point clearer in the revision, since one of the main takeaways is that the learned representations transfer well from AFDB-derived structures to experimentally resolved ones.

---

> > ### Author Rebuttal · Reviewer_7wxf · 2026-04-06
> >
> > Thank you for addressing many of the concerns raised.
> >
> > My only concern is way paper claim about the benchmark. For clarity and to have user aware of the limits, the authors should mention that this dataset is focused on the classification of largest domain in the abstract and earlier in the description.

---

> > > ### Author Response · Authors · 2026-04-07
> > >
> > > Dear Reviewer 7wxf,
> > >
> > > Thank you for your positive follow-up and additional feedback. We will carefully incorporate the rebuttal and your additional feedback into the revision.
> > >
> > > Best regards,
> > >
> > > The Authors

---

### Official Review · Reviewer_jSdk · 2026-03-15

**Soundness:** 3
**Presentation:** 3
**Significance:** 3
**Originality:** 3
**Overall Recommendation:** 5
**Confidence:** 3

**Summary:**

The authors of this paper introduce TEDBench, a large-scale benchmark for protein fold classification derived from the TED (Encyclopedia of Domains) resource and Foldseek-clustered AlphaFold structures. The dataset contains 462k predicted structures with labels and 27k experimental structures as an external test set and covers 965 topology classes, addressing redundancy and scale limitations present in prior datasets. They furthermore propose Masked Invariant Autoencoders (MiAE), a self-supervised framework for protein structure representation learning, that achieves competitive performance on TEDBench.

**Compliance With Llm Reviewing Policy:**

Affirmed.

**Final Justification:**

I would like to thank the authors for their rebuttal and the new experiments. Overall, the rebuttal resolves my main concerns, improves the paper’s framing, and makes me more positive about the submission. I am therefore increasing my score.

**Key Questions For Authors:**

1. How does performance scale with the amount of pretraining data? Could the authors provide an ablation study or establish scaling laws?

**Limitations:**

yes

**Strengths And Weaknesses:**

## Strengths

1. The authors created TEDBench, a large and carefully curated dataset for protein fold classification. The dataset contains over 460k predicted protein structures and 27k experimentally determined structures, which will be useful for the community.

2. The authors provide a comprehensive analysis of various baseline types (equivariant GNNs, sequence models, structure models, and hybrid models). They also include ablation studies for various aspects of the proposed architecture (masking ratio, decoder depth, and width, etc.), enabling the reader to understand more thoroughly the architectural choices.

## Weaknesses

1. The paper positions itself as both a benchmark paper and a method paper, but the focus between these two contributions is a bit unclear. Firstly, the introduction and motivation emphasize TEDBench as a large-scale dataset intended to fill an important gap in protein fold classification benchmarks. On the other hand, a significant portion of the paper is devoted to introducing and evaluating the MiAE model. Therefore, this makes it difficult to fully determine whether the central contribution is the benchmark itself or the proposed method.

---

> ### Author Rebuttal · Authors · 2026-03-30
>
> We appreciate the reviewer’s recognition of TEDBench as a carefully curated dataset and our comprehensive baseline analysis. Below, we address the points they raised.
>
> > W1. Positioning of the paper
>
> We thank the reviewer for this comment and agree that the current framing may make the paper seem split between two contributions. Our main contribution is TEDBench as a large-scale benchmark for protein fold classification. At the same time, we believe MiAE is an important methodological contribution **for this specific task**, since part of the goal of the paper is to understand what kind of training recipe is effective for fold classification at this scale.
>
> We will revise the introduction and contributions section to make this relationship clearer. Concretely, we will present TEDBench as the primary contribution, and MiAE as a task-motivated method that both provides a strong reference baseline and helps study how protein structure models can be trained effectively for large-scale fold classification. We will also add a brief compute comparison to motivate this point:
>
> | Model       | Pre-training (GPU hours) | Fine-tuning (GPU hours) |
> | ----------- | ------------------------ | ----------------------- |
> | ESM2-650M   | 138,240                  | 132                     |
> | SaProt-650M | 138,240                  | 132                     |
> | MiAE        | 768                      | 43                      |
>
> Note that the architectures of SaProt and ESM2 and the sequence lengths are identical. The pre-training time of ESM2 and SaProt is the same and was estimated based on the numbers reported in the SaProt paper (Su et al., ICLR 2024), while the fine-tuning results were obtained from our compute cluster using the same hardware for all models. Our comparison suggests that MiAE requires much fewer computational resources to achieve similar or better performance than state-of-the-art protein representation learning models such as ESM2 or SaProt.
>
> We believe this framing better reflects the paper: **TEDBench is the central contribution, while MiAE shows a practical and effective way to approach this task**.
>
> Su et al., SaProt: Protein Language Modeling with Structure-aware Vocabulary, ICLR 2024.
>
> > Q1. Scaling with Pretraining Data
>
> Thank you for the suggestion. We performed this ablation and will add the corresponding plots to the revised paper. We also provide the figure in this [anonymous link](https://figshare.com/s/ecb48c4b4f7d76cf0521). In both linear probing and fine-tuning, performance improves consistently as the amount of pretraining data increases, on both the test and the external test set. For example, in linear probing, Macro F1 improves from about 27 to 59 on the test set and from 38 to 66 on the external test set as pretraining data scales from roughly $10^4$ to $10^6$ proteins. Fine-tuning shows the same trend, improving from about 68 to 74 on the test set and from 74 to 76 on the external test set.
>
> We agree that this is an important analysis. While these results do not establish a formal scaling law, they do show a clear positive scaling trend with more pretraining data. We note that the fine-tuning experiments here were trained with a layer-wise learning rate decay and fewer epochs, optimized for the MAE paradigm, which could lead to suboptimal convergence and perform worse than the models trained from scratch.

---

> > ### Author Rebuttal · Reviewer_jSdk · 2026-04-04
> >
> > I would like to thank the authors for their rebuttal and the new experiments. Overall, the rebuttal resolves my main concerns, improves the paper’s framing, and makes me more positive about the submission. I am therefore increasing my score.

---

> > > ### Author Response · Authors · 2026-04-06
> > >
> > > Dear Reviewer jSdk,
> > >
> > > We sincerely appreciate your positive follow-up and your confirmation that our rebuttal has addressed your main concerns. We will carefully incorporate the discussed improvements into the revision.
> > >
> > > Best regards,
> > >
> > > The Authors

---

### Decision · Program_Chairs · 2026-04-30

**Decision:**

Accept (spotlight)

**Comment:**

The authors created TEDBench, a large and carefully curated dataset for protein fold classification. The dataset contains over 460k predicted protein structures and 27k experimentally determined structures, which will be useful for the community. The reviewers agreed that the dataset addresses an important problem, has rigorous splits, and that the experiments are comprehensive and informative. The paper is generally well-organized and clearly written, making it easy to follow the methodology and results. The supplemental material is particularly helpful, containing further details on training procedures and additional results.

There is one outstanding concern about the correctness of only including one domain from multi-domain proteins, but the reviewers agree that this can be addressed by making it more transparent in the text.